# Combined Modality Bladder-Sparing Therapy for Muscle-Invasive Bladder Cancer: How (Should) We Do It? A Narrative Review

**DOI:** 10.3390/jcm12041560

**Published:** 2023-02-16

**Authors:** Artur Lemiński, Wojciech Michalski, Bartłomiej Masojć, Krystian Kaczmarek, Bartosz Małkiewicz, Jakub Kienitz, Barbara Zawisza-Lemińska, Michał Falco, Marcin Słojewski

**Affiliations:** 1Department of Urology and Urological Oncology, Pomeranian Medical University, 70-111 Szczecin, Poland; 2Department of Urological Cancer, Maria Sklodowska-Curie National Research Institute of Oncology (MSCNRIO), Roentgena 5, 02-781 Warsaw, Poland; 3Department of Radiotherapy, West-Pomeranian Oncology Center, 71-730 Szczecin, Poland; 4Department of Minimally Invasive and Robotic Urology, University Center of Excellence in Urology, Wroclaw Medical University, Borowska 213, 50-556 Wrocław, Poland; 5Department of Rehabilitation, West-Pomeranian Oncology Center, 71-730 Szczecin, Poland

**Keywords:** muscle-invasive bladder cancer, bladder-sparing treatment, radical cystectomy, radiotherapy, neoadjuvant chemotherapy

## Abstract

Organ-sparing combined-modality treatment for muscle-invasive bladder cancer remains underutilized despite high-quality evidence regarding its efficacy, safety, and preservation of quality of life. It may be offered to patients unwilling to undergo radical cystectomy, as well as those unfit for neoadjuvant chemotherapy and surgery. The treatment plan should be tailored to each patient’s characteristics, with more intensive protocols offered to patients who are fit for surgery but opt for organ-sparing. After a thorough, debulking transurethral resection of the tumor and neoadjuvant chemotherapy, the response evaluation should trigger further management with either chemoradiation or early cystectomy in non-responders. A hypofractionated, continuous radiotherapy regimen of 55 Gy in 20 fractions with concurrent radiosensitizing chemotherapy with gemcitabine, cisplatin, or 5-fluorouracil and mitomycin C is currently preferred based on clinical trials. The response should be evaluated with repeated transurethral resections of the tumor bed and abdominopelvic-computed tomography after chemoradiation, with quarterly assessments during the first year. Salvage cystectomy should be offered to patients fit for surgery who failed to respond to treatment or developed a muscle-invasive recurrence. Non-muscle-invasive bladder recurrences and upper tract tumors should be treated in line with guidelines for respective primary tumors. Multiparametric magnetic resonance can be used for tumor staging and response monitoring, as it may distinguish disease recurrence from treatment-induced inflammation and fibrosis.

## 1. Introduction

Muscle-invasive bladder cancer (MIBC) is a challenging malignancy, requiring a prompt diagnostic work-up and a multidisciplinary approach to treatment. Radical cystectomy (RC) with pelvic lymph node dissection has been considered the gold standard treatment for MIBC for several decades, and although still widely employed, it has become embedded in a multidisciplinary treatment protocol, which includes administration of neoadjuvant cisplatin-based combination chemotherapy (NAC) in eligible patients [1,2,3]. Despite established efficacy and long-term oncological outcomes, RC remains a highly morbid operation, associated with significant rates of complications, regardless of the surgical approach utilized [4,5,6,7]. The introduction of laparoscopic and robotic RC along with standardized perioperative pathways, such as enhanced recovery after surgery (ERAS), has led to a modest decline in early morbidity, mortality, and postoperative infections; however, Psychological burden in RC patients may not be limited to those with non-orthotopic urinary diversion [8,9,10,11,12]. Moreover, the increasing life expectancy and aging of Western societies have led to a higher proportion of elderly patients with MIBC, which, given their increased comorbidity and risk of frailty, makes surgery an even more challenging choice [13]. Although advances in surgical techniques and perioperative care allow most of the elderly to undergo RC with acceptable perioperative risks, a substantial proportion of these patients eventually die within a year of the surgery [14]. Consequently, in progressively aging populations, there is a growing number of patients with MIBC who are considered unfit for RC. A significant part of this population receives nonradical therapies with detrimental influence on survival [15,16,17]. This phenomenon emphasizes the growing demand for alternative treatments of MIBC that would carry a lower risk of adverse outcomes and mortality, maintain or improve quality of life, and provide non-inferior oncological outcomes to RC. Bladder-sparing combined modality treatment (CMT), which involves maximal transurethral resection of tumor (TURBT), followed by NAC and chemoradiation (CRT) is currently employed as a curative alternative to RC.

Based on the available evidence, contemporary guidelines from the European Association of Urology (EAU) and American Association of Urology, in line with guidelines from several oncological associations, acknowledge the use of CMT in carefully selected, informed, and motivated patients with MIBC, who are focused on bladder preservation, and in those patients, who are considered unfit for radical cystectomy [1,17,18]. Despite this evidence-based endorsement, CMT remains underutilized, with only 10 to 15% of patients with MIBC receiving treatment with this modality [19,20].

With this narrative review, we update urologists on the current position of CMT in treatment of MIBC, discuss patient selection criteria for CMT, and propose patient pathways in which CMT should be considered.

## 2. Available Evidence on CMT

The efficacy and safety of chemoradiation for MIBC have been evaluated in the BC2001 prospective randomized controlled trial, which showed the advantage of radiotherapy (RT) combined with chemotherapy with 5-fluorouracil and mitomycin C (5-FU + MMC) over RT alone in terms of local disease control, without significant increase in long-term toxicity or deterioration in the quality of life [21,22]. Encouraging outcomes have also been reported from a phase II trial on hypofractionated RT with concurrent gemcitabine, with a 3-year overall survival (OS) of 75% and disease-specific survival (DSS) of 82% [23]. Several CMT strategies have been prospectively evaluated in Radiation Therapy Oncology Group (RTOG) trials, including phase II RTOG 88-02, 95-06, 97-06, 02-33, and phase III RTOG 89-03. The pooled data analysis from these trials by Mak et al. showed favorable results with 5- and 10-year DSS after CMT of 71 and 65% respectively, comparable with outcomes of immediate RC of similarly staged MIBC [4,7,24,25]. Similarly, a propensity-score matched single center evaluation of RC and CMT by Kulkarni et al. showed equivalent 5-year DSS rates of 73.2% and 76.6% for each modality, respectively [26]. Although there are no randomized, prospective head-to-head comparisons of CMT and RC published to date, there is a growing body of evidence originating from case series, matched comparisons, and meta-analyses showing similar oncological outcomes of RC and CMT [20,27,28,29]. The diversity of bladder-sparing strategies employed in clinical settings and the lack of a uniform definition of CMT have complicated several review papers that attempted to standardize and evaluate this approach. These include a systematic review by Ploussard et al., followed by a more recent systematic review and meta-analysis by Vashistha et al., who compared long-term outcomes of RC and CMT in a cohort of 9554 patients. The authors found no significant differences in 5- and 10-year OS and DSS between radical cystectomy and bladder preservation cohorts [30,31]. The largest systematic review and meta-analysis by Fahmy et al. compared the outcomes of RC and CMT in a cohort of over 30,000 patients, including 3402 patients who underwent CMT. The authors found complete responses (CR) after CMT in 75.3% of patients, with a mean 10-year DSS of 50.9%, compared to 57.8% after RC; *p* = 0.26 [19]. Furthermore, a recent multi-institutional-matched comparison of RC and CMT by Zlotta et al. conducted among 421 RC and 282 CMT patients suggested the advantage of bladder preservation approach in terms of DSS and OS: 85% vs. 78%, *p* = 0.02 and 78% vs. 70%, *p* < 0.001, for CMT and RC, respectively [32].

## 3. Potential Candidates

### 3.1. Patients Unwilling to Undergo RC

Planning treatment for patients with MIBC is a shared decision-making process, and patients should be made aware of all available options. Bladder-sparing CMT may be considered in patients who are fit for NAC and surgery but wish to avoid cystectomy to preserve quality of life. The majority of these patients present with good performance status and life expectancy; hence, it is vital to select those most likely to benefit from CMT as it may influence their prognosis. EAU guidelines, along with recommendations from several studies, emphasize the careful selection of patients in this setting, with optimal candidates having an organ-confined, clinically node-negative, preferably solitary cT2 tumor, less than 5 cm in diameter, without concomitant carcinoma in situ [1,25,26,31]. Patients also need to have a well-functioning bladder of adequate capacity and no hydronephrosis. There is an ongoing debate on the routine utilization of NAC in cisplatin-eligible patients who wish to undergo CMT [33,34,35]. The practice often adopted in clinical settings is to administer NAC to patients with MIBC who are fit for cisplatin and offer CMT to those who responded to NAC [36,37,38]. Similar selection criteria have been proposed in the SPARE Trial design (Figure 1) [39].

### 3.2. Patients Unfit for RC

Radiotherapy was historically employed as an alternative treatment for patients with MIBC who were considered unfit for RC. Chemoradiation alone or as a part of CMT may also serve this patient population, increasing the probability of response without incurring excessive toxicity [21,22]. Selection criteria in this cohort are usually less stringent; nonetheless, frail patients with small functional capacity bladders, severe lower urinary tract symptoms, and significant impairment of renal function are not likely to benefit.

## 4. Transurethral Resection

A thorough TURBT is typically the first step of treatment for imaging-detected bladder lesions, including MIBCs. At the time of initial TURBT, usually little is known about the patient’s preference regarding the treatment of eventual MIBC; therefore, whenever safely feasible, a maximal debulking of the tumor mass should be attempted. According to several reports from the Massachusetts General Hospital, achieving a visually complete resection is independently associated with the increased rate of complete responses after CRT, and improvements in OS, DSS, and bladder-intact disease-specific survival (BIS) stemming from lower odds of salvage cystectomy (SRC) [25,40]. In line with these findings, Suer et al. advocated a second TURBT, which has been shown to reveal residual tumor in over 60% of patients after visually complete initial resection and was associated with improved survival in patients undergoing CMT, compared to patients in whom no second resection was performed [41]. It remains undetermined whether the feasibility of a visually complete TURBT of MIBC is a prognostic factor per se, or represents a surrogate marker of a lower local stage of the tumor. Moreover, there is conflicting evidence regarding the influence of a visually complete initial resection on the response rate to NAC as seen in RC specimens, and on the long-term prognosis of patients who undergo NAC-RC [42,43]. Nonetheless, given the positive influence of complete TURBT on outcomes of CMT and largely unknown patient preferences at the time of the initial resection, it seems justified to perform a maximal safely achievable TURBT.

## 5. Neoadjuvant Chemotherapy

The MIBC is a highly chemosensitive tumor and therefore, cisplatin-based combination NAC became the standard of care before RC. The administration of NAC provides complete responses in one-third of patients and eradicates micrometastatic disease leading to lower odds of distant recurrence, improved local disease control, and long-term survival rates [2,44]. Whether the administration of NAC would bring a similar advantage to patients undergoing CMT has become a matter of debate [33,34,35,36]. The advanced bladder cancer (ABC) meta-analysis of 10 RCTs, including the data of 2688 patients, showed survival benefit of platinum-based combination NAC. The survival advantage from NAC was consistent, regardless of whether patients underwent RC, RT, or the combination of both as their local treatment for MIBC [45]. Similarly, an updated analysis of data from the BA06 RCT with a median follow-up of 8 years corroborated the OS advantage of 3 cycles of neoadjuvant cisplatin, methotrexate, and vinblastine (CMV) in both RC and RT arms. There was no evidence that the efficacy of CMV-based NAC differed in subgroups of patients choosing different forms of definitive local treatment. The allocation of local therapy was not random in this study, hence there were more favorable patient characteristics in the RC arm, which made the comparison of the RC and RT outcomes unfeasible [46]. These studies, although of high quality, have some drawbacks, i.e., patients were enrolled in the 1990s when contemporary NAC regimens were not yet developed. Moreover, the bladder-preserving arms enrolled high proportions of patients unfit for RC, and RT alone (rather than CRT) was used. More recently, retrospective data originated from the study by Jiang et al., who reported outcomes from their CMT protocol incorporating NAC [36]. After maximal TURBT, patients received NAC of gemcitabine and cisplatin (GC) for 2–4 cycles and were re-assessed on cystoscopy and computed tomography. Patients who progressed on treatment were scheduled for RC, while responders underwent conventionally fractionated RT of 60–66 Gy for bladder and pelvic lymph nodes with concurrent cisplatin at 40 mg/m^2^ weekly, for six weeks. Treatment was well tolerated with completion rates of 95% for NAC, 100% for RT, and 84% for at least 60% of the concurrent chemotherapy, respectively. Salvage cystectomy was required in 14% of patients and distant recurrence was diagnosed in 11% of patients, with a two-year BIS of 64% [36]. Conflicting evidence originates from the meta-analysis of Fahmy et al. in which the administration of NAC was not associated with a higher proportion of CRs with CMT, i.e., 76.2% for NAC–CMT vs. 73.0% for CMT alone; *p* = 0.33. Patients pretreated with NAC had similar survival rates to those who underwent CMT alone with 5-year OS and DSS in the NAC–CMT arm of 58.3% and 72.4% compared to 50.4% and 62.2% in CMT alone *p* = 0.078 and *p* = 0.13, respectively. Studies included in this meta-analysis were mostly retrospective, with very low utilization of NAC (3% and 13% for the RC and CMT groups, respectively); moreover, there was a substantial heterogeneity of CMT regimens, with CRT delivered either in a split-dose protocol with interim pathological assessment (30%) or as a continuous block of treatment (70%) [19]. These findings correspond with a retrospective National Cancer Database analysis by Royce et al., which included 2566 patients who underwent CMT due to cT2-4 N0 M0 MIBC. Eighteen percent of the study cohort received NAC. The study found no significant difference in 5-year OS rates, i.e., 30.6% for NAC-CMT vs. 31.8% for CMT, respectively. Interestingly, survival rates reported in this study were significantly lower than presented in the majority of literature, indicating a high proportion of patients unfit for other treatment options. In the multivariate analysis, NAC did not correlate with OS and no benefitting subgroups could be identified on sensitivity analyses [47].

In conclusion, there is high-quality evidence regarding the benefit of NAC in MIBC administered prior to RC, and prior to RT, but it is debatable whether this benefit extends to patients undergoing CMT. The available level 1 evidence originating from the BC2001 trial corroborated the feasibility and good tolerance of NAC with subsequent CRT or radiotherapy, but the study lacked the power to detect the eventual survival benefit associated with NAC in the CMT scenario [48]. Importantly, because of local disease downstaging, administration of NAC may render more patients with MIBC eligible for bladder sparing. The Vesper trial has proven that dose-dense MVAC (methotrexate, vinblastine, doxorubicin, cisplatin, i.e., ddMVAC) is superior to gemcitabine–cisplatin, showing that the former should be considered the regimen of choice for neoadjuvant treatment [49].

## 6. Chemo-Radiation

### 6.1. Radiotherapy

Radiotherapy has traditionally been an option for MIBC patients unfit for RC and chemotherapy. Patients with MIBC undergoing RT monotherapy experienced a CR rate of 64%, and a 5-year OS of 40%, respectively [50]. Plataniotis et al. summarized studies on RT schedules utilized for the treatment of MIBC over the last 20 years and compared them to more contemporary CRT schedules regarding rates of complete responses and overall survival. The average reported CR rates were significantly higher in studies reporting the efficacy of CRT, as compared to series involving RT alone at 75.9% and 64.4%, respectively. Studies on chemoradiation included two main types of regimens differing in response rates. Split-dose protocols with interim pathological assessment of the response after an induction CRT course, followed by a consolidation CRT in responders, had a mean CR rate of 71.6%, whereas continuous regimens, where a full dose of CRT was delivered, and the response was assessed after the completion of treatment had a mean CR rate of 79.6%. Immediate SRC was offered to non-responders from either treatment group [50].

Two fractionation regimens were employed to treat MIBC, i.e., the conventionally fractionated schedule of 64 Gy in 32 fractions over 6–7 weeks and the hypofractionated schedule of 55 Gy in 20 fractions delivered over 4 weeks. A recent meta-analysis of individual patient data compared these fractionation regimens in patients with high-risk non-muscle-invasive and muscle-invasive bladder cancers (T1G3 or T2–T4, N0M0) enrolled in two multicenter, randomized, phase-3 trials performed in the UK—BC2001 and British carbogen and nicotinamide (BCON) [21,51,52,53]. The study showed that patients who received 55 Gy in 20 fractions had a lower risk of invasive locoregional recurrence (adjusted HR 0.71), and similar toxicity profiles (adjusted RD–3.37%), compared to conventionally fractionated schedule. The authors of the meta-analysis concluded that 55 Gy in 20 fractions should be adopted as a standard of care for bladder preservation in patients with MIBC given its superiority to 64 Gy in 32 fractions with regard to invasive locoregional control and noninferiority with regard to toxicity [51].

### 6.2. Radiotherapy Target Definition

The BC2001 trial also investigated whether reducing the radiation dose to uninvolved portions of the bladder wall would diminish the side effects of treatment without impairing local control of the disease [54]. Fractionation schedules used in the study were either 55 Gy in 20 fractions over 4 weeks or 64 Gy in 32 fractions over 6.5 weeks. Two-hundred-and-nineteen patients were randomized to receive the standard whole-bladder RT or reduced high-dose volume radiation therapy (RHDVRT), which aimed to deliver a full radiation dose to the tumor and 80% of the maximum dose to the uninvolved bladder wall. The planning target volume (PTV) definition for patients allocated to the whole bladder RT (control) was the outer bladder wall plus the extravesical extent of the tumor with a 1.5 cm margin. For RHDVRT patients, two PTVs were defined, i.e., PTV1 for the previous group, and PTV2, which included gross tumor volume with the addition of a 1.5-cm margin. The study found no significant reduction of late side effects with RHDVRT, compared to the whole bladder RT as overall rates of significant toxicity were low and so were rates of the invasive relapse of cancer [54].

### 6.3. Choice of Radiosensitizing Agent

Several radiosensitizing strategies for CRT have been developed, primarily based on cytotoxic chemotherapy (Table 1).

Radiosensitizing chemotherapy regimens are often single-agent, and less dose-intensive than those utilized for systemic chemotherapy of MIBC; therefore, even patients not suitable for cisplatin-based combination NAC may still be eligible for CRT [23]. Nevertheless, single-agent chemotherapy concurrent to RT should not be considered a reliable systemic treatment from the perspective of eradication of micrometastases [35]. There is limited evidence supporting an ‘optimal’ choice of a radiosensitizer, as these regimens have not been directly compared in the setting of a randomized trial, and in available studies, they were combined with diverse radiation protocols with or without NAC. However, owing to similar outcomes obtained with different chemotherapy protocols, a personalized choice of a radiosensitizing chemotherapy regimen reflecting the patient’s condition, comorbidities, physician experience, and feasibility of treatment delivery is preferred. Single-agent weekly cisplatin or gemcitabine seem to be reasonable options for patients fit and unfit for cisplatin, respectively, along with a combination of 5-FU and MMC.

A promising method for improving RT efficacy was investigated in the BCON trial. Inhalation of carbogen (a mixture of gases containing 98% oxygen and 2% of carbon dioxide) with concurrent administration of nicotinamide (CON) was demonstrated to sensitize tumors to RT by increasing tissue oxygenation [53,58]. Combining CON with a conventional or hypofractionated RT regimen led to a 13% improvement in the 3-year OS, as compared to RT monotherapy [53]. The 10-year update of the BCON trial data revealed a comparable 10-year OS between CON-RT and RT monotherapy in the entire study population; nonetheless, the subgroup of patients with pathological features of tumor necrosis was found to benefit from the hypoxia-modifying approach [52].

### 6.4. Immunotherapy

Several early-phase studies investigated the efficacy and safety of immune checkpoint inhibition as a complement to existing CMT protocols. Agents investigated include atezolizumab, pembrolizumab, and nivolumab along with a combination of nivolumab and ipilimumab [59,60,61]. De Ruiter et al. reported on the addition of 480 mg of nivolumab (NIVO480) or a combination of 3 mg/kg of nivolumab and 1 mg/kg of ipilimumab (NIVO3 + IPI1) to CRT of 55 Gy in 20 fractions and 5-FU/capecitabine in patients with MIBC. In this dose-finding study, the authors found both regimens of acceptable toxicity, with 90% OS at 2 years for CRT + NIVO480 and 90% OS at 1 year for CRT/NIVO3 + IPI1. The third arm investigating CRT and nivolumab 1 mg/kg with ipilimumab 3 mg/kg was discontinued due to excessive toxicity [61]. The ongoing INTACT phase III randomized trial investigates concurrent and adjuvant atezolizumab with CRT versus CRT alone in a cohort of MIBC patients. The choice of the radiosensitizer is clinician-dependent and includes twice-weekly gemcitabine, weekly cisplatin, or 5-FU + MMC. The study recently reported a pre-specified safety analysis and found concurrent immunotherapy safe with an increase of grade 3 hematological toxicity in the immunotherapy arm, but no significant increase in gastrointestinal or genitourinary toxicity [60].

### 6.5. Chemoradiation in Elderly Patients

There is growing evidence regarding the efficacy and safety of CMT in elderly populations with MIBC. A retrospective study from Christodoulu et al. investigated the efficacy and tolerability of CRT with gemcitabine radiosensitization in patients over 75 years old, and compared the outcomes to patient data from the BCON phase III trial [62]. The CMT protocol in this study consisted of TURBT, followed by a platinum-based combination NAC, and CRT with weekly gemcitabine (100 mg/m^2^). The study showed that a similar proportion of elderly patients received a planned dose of RT, but fewer completed all four cycles of gemcitabine due to the increased toxicity. The DSS, progression-free survival, and local progression-free survival were not significantly affected by age in the multivariate analysis. Moreover, OS and local progression-free survival in elderly patients were comparable between BCON and GemX trials (HR 1.13, *p* = 0.616 and HR 0.85, *p* = 0.659 respectively). Based on these encouraging results, the authors concluded that radiosensitization with gemcitabine is safe and effective in elderly populations and should be considered for this group of patients with MIBC [62].

### 6.6. Quality of Life in Patients Treated with CRT

Few studies have analyzed the long-term toxicity following bladder preservation treatment for MIBC, and most of them are retrospective. The largest randomized study in this area, the BC2001, included prospective assessments of the patients’ reported outcomes. The evaluation of QoL data of BC2001 was reported in a recent paper by Huddart et al., showing a transient decline in health-related quality of life (HRQL) at the end of CRT, with a subsequent return to baseline at 6 months [22]. Moreover, the addition of chemotherapy (5-FU + MMC) to RT did not affect the reported HRQL outcomes. The study provided evidence that CRT with 5-FU + MMC improves locoregional control of MIBC without a significant decline in HRQL [22].

## 7. Follow-Up Regimen and Salvage Treatment

The follow-up after CMT is aimed at detecting bladder recurrences, distant treatment failures, and de novo tumors of the urothelial lining within the upper urinary tract and urethra. Patients after CMT undergo a close follow-up surveillance, including cystoscopic bladder evaluations and biopsies, along with abdominopelvic computed tomography (CT). Existing evidence is insufficient to make definitive recommendations on the optimal follow-up regimen. Sanchez et al. found that the median time to first non-muscle-invasive bladder cancer (NMIBC) recurrence was 1.8 years, and 82% of NMIBC relapses developed within the first 5 years of follow-up [63]. Moreover, data from the pooled analysis of six RTOG studies demonstrated that NMIBC recurrence rates were 31% and 36% at 5 and 10 years, respectively [24]. These patterns of recurrence support the follow-up scheme recommended by the National Comprehensive Cancer Network (NCCN) guidelines, which includes cystoscopic surveillance every 3 months for 2 years, every 6 months for years 3 to 5, and then yearly for life [64]. Endoscopic examinations are complemented with transurethral resections/biopsies of the tumor scar and bimanual examination under anesthesia (EUA) during the first six months of follow-up [25]. Repeated TURBTs allow the detection of a tumor recurrence underneath the bladder scar, which may go unnoticed during office cystoscopy. In patients with negative evaluations during the first 6 months of follow-up, tumor site resections and EUAs may be discontinued unless there were suspicious findings in the office cystoscopy [40,63]. Some recent studies have included evaluations of voided urine cytology, which was obtained before each cystoscopy; however, the cytological assessment of the irradiated urothelium was reported as unreliable [25,63,65].

CT of the thorax, abdomen, and pelvis is recommended for the follow-up to detect relapse in regional lymph nodes and distant metastasis after CMT with curative intent [66]. Imaging surveillance after CMT should include CT scans every 3–6 months for 2 years and every 12 months for 3–5 years [64]. An analysis of prospective RTOG protocols showed 5-year rates of regional nodal and distant recurrences of 12% and 32%, respectively. Meanwhile, 10-year rates slightly increased to 14% and 35%, respectively. Another CMT study that offered a long-term follow-up and addressed the risk of late metastasis suggested a flattening of the CSS curve beyond the first 5 years after completing CMT, similar to the follow-up after RC [40]. Therefore, routine CT imaging may be discontinued after five years in most patients [66] (Figure 1).

Multiparametric magnetic resonance imaging (mpMRI) is becoming increasingly utilized in the staging of bladder cancer owing to its greater reliability in the detection of muscle invasion and the presence of extravesical disease, along with the standardized vesical imaging-reporting and data system (VI-RADS) [67,68]. Moreover, mpMRI and VI-RADS may serve as diagnostic modalities in the treatment response evaluation, facilitating clinical decision-making in the context of bladder-sparing [69,70]. The role of mpMRI in response to monitoring post-CMT is less well-established; however, some authors infer the added benefit of diffusion-weighted sequences to discriminate disease recurrence from post-treatment fibrosis and inflammation within the bladder wall [71].

### 7.1. Bladder Recurrence

The identification of any suspicious bladder lesions during CMT follow-up should be further investigated using TURBT. All subsequent management decisions depend on the accurate TURBT pathology. Therefore, the resection should be meticulous and as complete as possible, including the use of second-look TURBT when indicated. Among patients with CR after CMT, the rates of NMIBC and MIBC recurrence vary between 24 and 29% and 12 and 16%, respectively [40,63,72]. Recurrent MIBC in patients fit for surgery should be managed with SRC. The evidence on the optimal management of recurrent NMIBC in a previously irradiated bladder is weaker. These patients may be managed with TURBT alone, TURBT and intravesical therapy (immunotherapy or chemotherapy), or SRC. In most previous studies, TURBT alone was the most common treatment for low-grade NMIBC recurrences, whereas recurrent high-grade NMIBC and carcinoma in situ were managed with TURBT and intravesical Bacillus Calmette-Guérin (BCG) therapy. Data from these studies emphasize that response rates to BCG therapy were similar to those seen in primary tumors [63,73,74,75]. Interestingly, although not yet studied in post-CMT patients, various BCG strains were found to perform differently in high-risk NMIBC, with several studies showing the superiority of the BCG-TICE strain over RIVM [76,77]. Despite adequate BCG exposure, Weiss et al. reported that 33% of patients had further NMIBC recurrences and 19% progressed to MIBC, respectively [75]. Moreover, DSS was not affected if NMIBC recurrence was treated with TURBT and intravesical therapy, as compared to SRC [63]. These results indicate that primary MIBC remains the key driver of disease-specific prognosis, rather than the choice of management of recurrent NMIBCs. Hence, CMT patients presenting with recurrent NMIBCs may avoid immediate SRC and should be further managed with the bladder-preserving approach. Unfortunately, despite comparable oncological outcomes of BCG therapy in the irradiated and non-irradiated bladder, the toxicity of intravesical therapy appears somewhat higher following CMT. There is an increased risk of bladder contracture and inability to tolerate normal 2 h dwell times of BCG instillation, presumably as a result of bladder shrinkage and bothersome storage symptoms [63,74,78,79]. Nevertheless, these side effects are still milder than those reported after SRC, which must be considered in the decision-making.

#### Salvage Cystectomy

Immediate cystectomy in patients counseled for CMT should be performed in scenarios in which the risk of metastatic spread increases. This means that RC should be carried out in patients who do not respond to adequately administered NAC. Salvage cystectomy should be performed if TURBT pathology reveals a residual or recurring ≥pT1 tumor after an induction CRT in a split-course scenario, or in the case of MIBC recurrence after completing CMT. In these situations, SRC remains the standard of care for patients who are fit for surgery. For carefully selected patients, the SRC approach improves oncological outcomes in comparison to the no SRC approach (whether patients are unfit for surgery or due to patient preferences) [80]. However, despite its clear oncological benefit, SRC still raises safety concerns in patients in whom CMT fails. Results from one of the largest cohorts of CMT patients from the Massachusetts General Hospital highlight that surgical complication rates are acceptable, albeit slightly higher than with upfront RC. The 90-day postoperative mortality and major complication rates were 2.2% and 16%, respectively. Cardiovascular and hematological complications were mostly observed after immediate SRC for an incomplete response during the split-course CRT, whereas SRC for recurrent MIBC is at higher risk of tissue healing complications [81]. Another concern regarding SRC is the appropriate selection of urinary diversion. There is a general reluctance to construct orthotopic neobladders after pelvic irradiation, and a common practice at many institutions is to utilize ureterostomy and ileal or transverse colon conduits rather than neobladders in the majority of SRC cases. This is justified by the higher risk of functional complications associated with neobladders in the irradiated pelvis. However, it is unknown whether induction chemoradiation at 40 Gy would affect the healing and function of neobladders. Considering that radiation fibrosis begins to occur 4–12 months after radiation therapy, the neobladder might be considered when the immediate SRC is performed after failure of induction CRT in a split-course scenario [82]. However, continuous CRT is currently the most common regimen used.

It is also necessary to raise the issue of management of MIBC recurrence in patients unfit for surgery. In general, further pelvic RT is not recommended after CMT due to the risk of significant toxicity from the cumulative radiation dose. Once the bladder is irradiated during CMT, further radiation heightens the risk of developing a contracted, non-functioning bladder as well as injury to adjacent organs. Considering the short life expectancy of these patients due to substantial comorbidities and significant concurrent mortality, the best supportive care and repeated, palliative TURBT to control bleeding and bothersome local symptoms may be preferred.

### 7.2. Non-Bladder Recurrence

The possibility of the upper urinary tract or urethral recurrences should be considered in patients treated with CMT. However, evidence in this area is sparse, and more studies are needed to determine recurrence rates in this population. Due to insufficient evidence, CMT patients with upper urinary tract or urethral recurrences should be managed similarly to patients with primary tumors in these locations. Meanwhile, the 10-year distant metastasis rates after CMT are roughly 30–35% [25,40]. In the study by Rödel et al., the 5-year metastasis-free survival was 79% in a subset of patients who achieved CR with CMT, but only 52% in those who failed to respond [80]. The treatment of patients with distant metastasis after CMT should follow current guidelines, with cisplatin-based chemotherapy as the first-line treatment option for eligible patients, and immunotherapy with checkpoint inhibitors for those unfit for cisplatin. Metastasectomy or stereotactic radiotherapy may be considered in selected cases with solitary metastatic deposits [66].

## 8. Conclusions

The combined modality bladder-sparing approach is here to stay. It offers a chance of cure for a significant proportion of patients with MIBC and preservation of quality of life. CMT may be offered to clinically diverse groups of patients, and the treatment plan needs to be tailored to the patient’s characteristics. Importantly, in patients fit for cisplatin, the choice of local treatment for MIBC should be made after the completion of NAC, as the response to systemic treatment may render more patients eligible for organ-sparing. The radiotherapy of choice is a hypofractionated regimen of 55 Gy in 20 fractions delivered as a continuous block of therapy, combined with concurrent chemotherapy with gemcitabine, cisplatin, or 5-FU + MMC. Disease recurrences should be detected during follow-up studies, aimed at local relapse, metastasis, or de novo occurrence of disease in the upper urinary tract. Due to the lack of specific studies for patients after CMT, non-muscle-invasive bladder recurrences and upper tract tumors diagnosed during the follow-up should be treated according to the available guidelines for respective primary tumors. A salvage cystectomy is required in patients who are fit for surgery if MIBC recurs. Prospects of CMT include the incorporation of multiparametric magnetic resonance imaging in staging, treatment allocation, and response monitoring, as well as a broader utilization of concurrent and adjuvant immunotherapy after the completion of ongoing clinical trials.

## Figures and Tables

**Figure 1 jcm-12-01560-f001:**
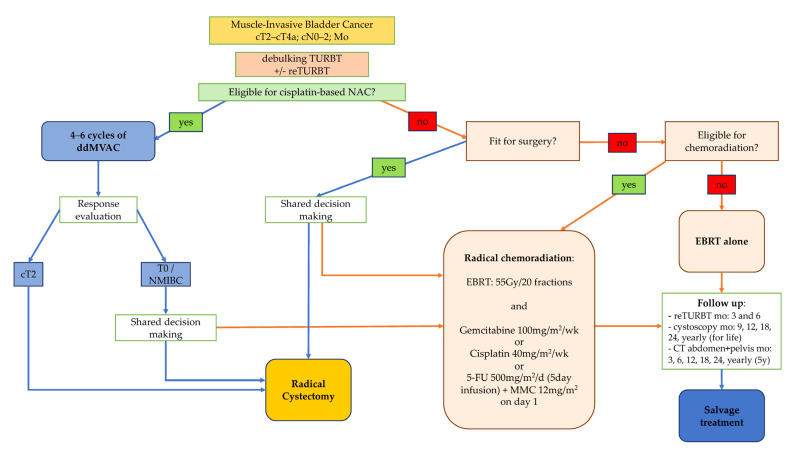
Flowchart illustrating the proposed treatment planning process in patients with muscle-invasive bladder cancer, incorporating CMT. CT: computed tomography; ddMVAC: dose-dense methotrexate, vinblastine, doxorubicin, cisplatin; EBRT: external beam radiation therapy; NAC: neoadjuvant chemotherapy; NMIBC: non-muscle-invasive bladder cancer; TURBT: transurethral resection of bladder tumor.

**Table 1 jcm-12-01560-t001:** Cytotoxic agents commonly used for CRT.

Agent(s)	Dosage	Regimen	Trial	Reference
Cisplatin	40 mg/m^2^	weekly	-	[36]
	70 mg/m^2^	three-weekly	RTOG 8802	[24]
	100 mg/m^2^	three-weekly	RTOG 8903	[24]
Gemcitabine	100 mg/m^2^	weekly	GemX	[23]
Mitomycin C + 5-fluorouracil	12 mg/m^2^ 500 mg/m^2^	day 1 days 1–5; 16–20	BC2001	[21]
Mitomycin C + Capecitabine	12 mg/m^2^ 750–825 mg/m^2^	day 1 TID excl. weekends	-	[55]
Cisplatin + 5-fluorouracil	15 mg/m^2^ 400 mg/m^2^	days 1–3; 8–10; 15–17	RTOG 0233	[56]
Cisplatin + Paclitaxel	20 mg/m^2^	days 1–2; 8–9; 15–16	RTOG 9906 RTOG 0233	[57] [56]
	50 mg/m^2^	weekly		

TID–twice daily; RTOG–Radiation Therapy Oncology Group.

## Data Availability

Not applicable.

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
