# Peer review of "Combined Modality Bladder-Sparing Therapy for Muscle-Invasive Bladder Cancer: How (Should) We Do It? A Narrative Review"

_jcm, 2023, doi:10.3390/jcm12041560_

Round 1
Reviewer 1 Report
Authors should be congratulated for addressing an important topic on growing demand for alternative treatment of MIBC. The goal should be finding new strategies to carry lower risk of adverse outcomes and mortality, maintain, or improve quality of life and provide a non-inferior oncological outcome to RC. The review needs some restructuration and some amendments are required.
1. pp. 53 "nearly half of these patients eventually die within one year of surgery". this sentence is disproportionate, please eliminate it or express the concept in milder terms.
2. I strongly recommend restructuring the introduction. When presenting several CMT strategies that have been prospectively evaluated, please do not report results and details to the discussion. Moreover, I recommend giving brief information on BC in risk factors. for the scope please cite this recent paper on meat intake risk factors BC associated (DOI: 10.3390/cancers14194775).
3. What is the author's position on partial cystectomy?
4. You mentioned several papers on trimodal therapy but you did not address the topic in a systemic way. May consider filling a dedicated paragraph.
5. I recommend emphasizing BCG therapy as the standard of care in the NMIBC. please include this citation which compares the adjuvant induction ± maintenance setting of intravesical immunotherapy with either BCG TICE or RIVM (DOI: 10.3390/cancers14040887).
6. Check typos.
Reviewer 2 Report
I thank the Editors for the possibility to review this interesting narrative review concerning bladder-sparing trimodality treatment for patients with MIBC.
The topic is interesting indeed and the manuscript is well written.
From a methodological point of view, a systematic review could sound more interesting to the reader.
Though, several systematic review are already available concerning this topic: must be cited in the present manuscript and the authors should discuss the advantages of their manuscript compared to the already available evidences.
However, the present manuscript could be re-considered for publication after major revision:
1) Potential benefit of a Robot assisted approach are not discussed in the introduction section of the present paper. Please expand this topic. You may consider citing:
PMID: 33712389. PMID: 32847113. PMID: 31190152. PMID: 29281852
Round 2
Reviewer 1 Report
The authors provided a deeply improved revised version of the manuscript. I believe has been sufficiently improved although I disagree with the exclusion of the suggested paper.
Reviewer 2 Report
Manuscript suitable for publication